# Proceedings of the Online Conference “Vaccines and Vaccination during and Post COVID Pandemics” (7–9 December 2022)

**DOI:** 10.3390/vaccines11071175

**Published:** 2023-06-29

**Authors:** Liba Sokolovska, Maria Isaguliants, Franco M. Buonaguro

**Affiliations:** 1Institute of Microbiology and Virology, Riga Stradins University, LV-1007 Riga, Latvia; 2Experimental Oncology Department, National Cancer Institute ‘Fondazione Pascale’, 80131 Naples, Italy; f.buonaguro@istitutotumori.na.it

**Keywords:** vaccines, adjuvants, COVID-19 pandemic, HIV-1, high-risk HPV, HBV, cancer, cancer vaccine, vaccine hesitancy

## Abstract

The COVID-19 pandemic put focus on various aspects of vaccine research and development. These include mass vaccination strategies, vaccination compliance and hesitancy, acceptance of novel vaccine approaches, preclinical and animal models used to assess vaccine safety and efficacy, and many other related issues. These issues were addressed by the international online conference “Vaccines and Vaccination During and Post COVID Pandemics” (VAC&VAC 2022) held on the platform of Riga Stradins University, Riga, Latvia. Conference was supported by the International Society for Vaccines, the National Cancer Institute “Fondazione Pascale” (Naples, Italy), and the scientific journal VACCINES (mdpi). VAC&VAC 2022 attracted nearly 150 participants from 14 countries. This report summarizes conference presentations and their discussion. Sessions covered the topics of (1) COVID-19 vaccine development, evaluation, and attitude towards these vaccines, (2) HPV and cancer vaccines, (3) progress and challenges of HIV vaccine development, (4) new and re-emerging infectious threats, and (5) novel vaccine vehicles, adjuvants, and carriers. Each session was introduced by a plenary lecture from renowned experts from leading research institutions worldwide. The conference also included sessions on research funding and grant writing and an early career researcher contest in which the winners received monetary awards and a chance to publish their results free of charge in the special issue of VACCINES covering the meeting.

## 1. Online Conference “Vaccines and Vaccination during and Post COVID Pandemics” (7–9 December 2022)

The emergence of SARS-CoV-2 and the following COVID-19 pandemic created a significant public health crisis. To mitigate the spread of SARS-CoV-2 and the disease burden of COVID-19, the development of an efficacious vaccine was crucial. With the rapid development, testing, and implementation of such a vaccine, many aspects of vaccine research and development were put into the focus of both the public and the scientific community. Such issues as mass vaccination strategies, vaccination compliance and hesitancy, acceptance of novel vaccine approaches, preclinical and animal models used to assess vaccine safety and efficacy, and many others were broadly pondered and discussed and were also covered during the “Vaccines and Vaccination During and Post COVID Pandemics” online conference (7–9 December 2022).

The online conference VAC&VAC 2022 gathered about 150 participants from Latvia, Sweden, Italy, the USA, South Korea, South Africa, Netherlands, China, India, Tanzania, Denmark, Germany, Lithuania, and San Marino. Among the participants were academic researchers, public health specialists, and industry representatives, and one-fourth were medical students and PhD students. The conference was supported by the Latvian Council of Science grant 2021/1-0484, Riga Stradins University, the National Cancer Institute “Fondazione Pascale” (Naples, Italy), the International Society for Vaccines (https://isv-online.org/, accessed on 27 June 2023), and the MDPI Journal *Vaccines*.

The conference covered six main topics—COVID-19 Pandemics & Vaccines, Development of HIV vaccines, HPV infection & HPV vaccination, Cancer Vaccines, Facing New and Re-Emerging Threats, Novel Vaccine Vehicles, Adjuvants, and Carriers, with additional sessions on Research Funding, Impact, and Implementation. Sessions included 32 oral presentations opened by plenary lectures by experts in the respective fields. Presentations and the book of abstracts are available on the conference website https://www.rsu.lv/en/vac-vac-2022 (accessed on 27 June 2023). Conference participants were invited to contribute their research findings to the special VACCINES (MDPI) issue, “Pandemics-Born Revolution in the Preclinical and Clinical Trials of Microbial Vaccines”.

Early career researchers presenting their data at the conference participated in the contest for the best study. Their presentations, seven in total, were evaluated by the board of five independent reviewers—one professor, one associated professor, two senior researchers, and one PhD student, with nominations of 1st, 2nd, and 3rd place winners.

## 2. Session 1: COVID-19 Pandemics & Vaccines

The conference was opened by a session on COVID-19, covering disease pathogenesis, vaccine development, vaccine-induced immune responses, and public attitude toward vaccination. The session was co-chaired by Prof. Franco M. Buonaguro and Dr. Maria Isaguliants.

The first plenary lecture was given by Prof. **Ranjit Ray** (Saint Louis University, Saint Louis, MO, USA) on SARS-CoV-2 and inflammation. Prof. Ray is an expert in RNA viruses, especially in the molecular mechanisms of virus-host interactions, pathogenesis, and vaccine development. The lecture outlined the clinical course of COVID-19 and the inflammatory response induced during the disease. The clinical course of COVID-19 is extremely variable, and the disease outcome is heavily reliant on the capabilities of the immune system of the host. The inflammatory response plays a significant role in the pathogenesis of COVID-19, with cytokine storm being a hallmark of the disease. His group aimed to understand the mechanism(s) driving SARS-CoV-2-induced inflammatory response in experimental studies modeling the interaction between SARS-CoV-2 and human epithelial cells and their contribution to the cytokine storm. The studies demonstrated that SARS-CoV-2 inhibits ACE-2 expression, leading to an increased expression of angiotensin I, which in turn can lead to the synthesis and release of IL-6 and IL-6 trans-signaling (via the soluble IL-6 receptor) and subsequent hyperinflammatory response.

Additionally, they demonstrated that SARS-CoV-2 spike protein stimulates IL-6 and soluble IL-6R production and activates the transcription factors for IL-6 synthesis (MAPK and ADAM-17). They also demonstrated that the activation of IL-6 trans-signaling in the SARS-CoV-2 spike protein expressing epithelial cells forms an inflammatory circuit in the endothelial cells. Their studies revealed that spike-expressing epithelial cells release alarmins that can cause uncontrolled inflammatory activity near the secretory cells and at a distance [1]. Further studies of the epithelial cells revealed that the expression of SARS-CoV-2 spike protein triggers a localized form of endothelial senescence, resulting in enhanced leukocyte adhesion and the development of microvascular complications [2]. The authors observed that exosomes obtained from the plasma of COVID-19 patients contain both pro-inflammatory molecules [3] and the viral spike protein. Thus exosomes appear to be important in multiorgan pathogenesis during virus infection. These findings shed light on specific pathways by which the SARS-CoV-2 and its spike protein cause systemic inflammatory effects in individuals diagnosed with COVID-19.

The opening lecture was followed by an overview of DNA vaccines developed against SARS-CoV-2 by Dr. **Joel Maslow** from GeneOne Life Science Inc. (Seoul, Republic of Korea), which focuses on nucleic acid-based biomedicines including vaccines, plasmid-encoded monoclonal antibodies, and recombinant proteins. Dr. Maslow is currently the Chief Medical Officer of GeneOne and is leading the development of nucleic acid-based vaccines against emerging infectious diseases. Dr. Maslow reviewed the DNA vaccines created against SARS-CoV-2, their efficacy in eliciting both humoral and cellular immune responses, and results from the studies where these vaccines were used as boosters following mRNA vaccination. The authors discussed three DNA vaccines in more detail: ZyCoV-D, approved for use in India; INO-4800, which has concluded its phase 3 clinical trial; and GLS-5310, currently in phase 2 clinical trial. The results of the ZyCoV-D phase 3 trial demonstrated the induction of seroconversion and an efficacy of 66.6% as measured by the prevention of symptomatic infection [4]. All three vaccines elicited comparable end-point levels of binding and neutralizing antibody responses, and GLS-5310 also induced the most robust specific T-cell response. This talk presented DNA vaccines as an intriguing platform for future outbreaks as the trials demonstrated increased T-cell immune responses. DNA vaccines have several advantages, such as relatively fast production and stable storage conditions. Dr. Maslow also discussed the possibilities of noninvasive delivery of DNA vaccines by sonoporation.

In the next talk, Dr. **Manki Song** (International Vaccine Institute, Seoul, Republic of Korea) presented the development of viral vaccines, specific vaccines against SARS-CoV-2, on the platform of Vesicular Stomatitis Virus (VSV). The talk introduced the VSV vector and its main advantages—the absence of antibodies against the virus in the population and its mild pathogenicity for humans. To make it safer but still immunogenic, the vector was attenuated by introducing M gene mutations which significantly reduced the burst size of the virus without affecting the level of protein expression [5]. Dr. Song presented examples of the recombinant vector used in an Ebola vaccine and the work done to develop COVID-19 vaccines using this vector. The presentation also reported on the studies exploring recombinant VSV vaccine candidates containing different SARS-CoV-2 protein-coding sequences: full-length spike protein, the S1 subunit or the receptor binding domain of the spike protein, and the envelope protein [6]. The vaccine candidate containing full-length spike protein modified with melittin signal peptide at the N terminus and VSV G protein transmembrane domain and the cytoplasmic tail at the C terminus showed the most promising results in mice in terms of the high levels of binding and neutralizing antibodies and the protective effect after viral challenge.

Following the theme of innovative vaccine solutions Prof. **Edward Rybicki** (University of Cape Town, Cape Town, South Africa) gave a talk on the potential of plants as rapid-response expression platforms in the context of rapid vaccine development and diagnostic reagent production for pandemic response [7] with examples of SARS-CoV-2 and other (re)emerging viral threats. Prof. Rybicki’s main field of research is vaccine biotechnology, specifically the development of plant expression systems as platforms for the production of proteins. Multiple advantages of the plant-based systems were highlighted, including the lower costs of biomass production, less expensive infrastructure, more rapid scalability, and support of eukaryotic post-transcriptional modifications. Examples of real-life applications of the plant-based systems for vaccine production included plant-produced virus-like particle vaccine for influenza, which has already undergone Phase 3 trials, and the development of plant-produced vaccines for SARS-CoV-2 and West Nile virus. To conclude, Prof. Rybicki highlighted the potential of molecular plant farming for the fast and scalable production of reagents used for diagnostic tools and rapid-response vaccines.

Dr. **Zane Lucāne** (Riga Stradins University, Riga, Latvia) presented her findings on the long-term humoral and cellular immune responses elicited by SARS-CoV-2 vaccination in patients with primary antibody deficiencies (PADs). These conditions are variable and can present in substantially lowered levels of one or more of immunoglobulin isotypes—IgA, IgM, or IgG, e.g., in the case of selective IgA deficiency, IgA levels can be over 30 times lower than in the general population [8,9]. Patients suffering from PADs often present with frequent infections, and vaccination in some of them can prove to be ineffective [10]. This study included the immune response analysis of 28 patients suffering from PADs, either selective IgA deficiency or common variable immunodeficiency, previously vaccinated against SARS-CoV-2. This study demonstrated that in patients with primary antibody deficiencies, markers of humoral and cellular immune responses directed against SARS-CoV-2 spike protein persisted for up to a year following vaccination. Notably, even though the median anti-spike IgG antibody levels were significantly lower in the study group, the T-cell response was unchanged compared to the healthy controls.

Continuing the topic of COVID-19 vaccination, the two following talks focused on the public’s attitudes toward it. The first, by Dr. **Uldis Vēgners** (Riga Stradins University, Riga, Latvia), reported the findings of a study, “Hesitant bodies: a phenomenological analysis of the embodied experience of COVID-19 vaccine hesitancy”, aimed at understanding the basis of vaccine hesitancy. The preliminary results indicated that vaccine hesitancy should be considered not only as scepticism towards the safety and effectiveness of the specific vaccine but also as a broader reflection of an individual’s embodied connection with the world. The second talk continuing the topic, explored parents’ attitudes towards routine immunization of children in Belarus during the COVID-19 pandemic and was presented by Ms. **Yuliya Raukach** (Riga Stradins University, Riga, Latvia). By analyzing data acquired from 427 questionnaires, this study dissected parents’ attitudes towards routine vaccination, which sources of information about vaccination are used by parents, and whether their stance on vaccination was influenced or affected by the COVID-19 pandemic. The study found that most parents taking part in the questionnaire had a favorable attitude towards vaccination and that it was not significantly influenced by the pandemic. Yet, the results revealed that informing parents requires more attention and action. Studies by Dr. Lucāne and Ms. Raukach shared the 3rd prize of the conference as the best presentations.

## 3. Session 2: Development of HIV Vaccines

The second session of the conference covered several aspects of HIV vaccine research—the current progress and new promising approaches, vaccine trials, and therapeutic HIV vaccines. The session was chaired by Prof. **Jean Louis Excler** (International Vaccine Institute, Republic of Korea).

The session was opened with a plenary lecture on the current approaches to HIV vaccine development by Prof. **Jean Louis Excler** (International Vaccine Institute, Seoul, Republic of Korea). Prof. Excler has years of experience in vaccine development and immunization strategies. The lecture highlighted the challenges that HIV vaccine development faces, such as the genetic diversity of HIV-1, difficulty in reaching conserved epitopes of the virus, poorly immunogenic glycan shield, rare and slow development of neutralizing antibodies, the limited value of existing animal models, and finally the costly and lengthy efficacy trials which are challenged by the evolving prevention landscape [11]. Even though several HIV vaccine trials have failed, the research is still ongoing as the impact of an efficacious vaccine would be extraordinarily significant. A core issue of HIV vaccine development—eliciting broadly neutralizing antibodies (bNAb)—was addressed in more detail. Generally, three strategies to produce these antibodies are being explored in HIV vaccine research. They all involve the guidance of immune response maturation via sequential vaccination with different antigens. These strategies include (1) B-cell lineage vaccine design, where immunogens are derived from Env variants determined from analysis of persons with natural HIV-1 infection; (2) germline-targeting vaccine design, where the priming immunogens can induce the precursors of the bNAbs [12], and immunogens used for boosting are progressively more like the native Env, and (3) epitope-focused vaccine design where a series of immunogens aims to direct immune responses to target one or more structural epitopes of the Env [13]. The lecture then covered ongoing trials and promising new approaches, like combining vaccine candidates with pre-exposure prophylaxis (PrEP)—antiretroviral medication to protect individuals with high HIV-acquisition risk. This approach is being utilized in the African-led, European-supported PrEPVacc—an HIV prevention study running in East and Southern Africa from 2018 to 2024 (https://www.prepvacc.org/ accessed on 27 June 2023). This study simultaneously performs trials for two HIV vaccine candidates and a new PrEP medication. New vaccine vectors were also discussed, like the potential use of attenuated CMV as a vector. In an SIV model, the attenuated CMV vector-based vaccine demonstrated 80% protection from a second infection even after approximately three years following the initial challenge [14]. Finally, the lecture questioned whether the recent strides in DNA and mRNA vaccine development are the “Boost” and the “Accelerator” that HIV vaccine development needs. The experience and success of the vaccines developed for SARS-CoV-2 have given hope for developing an efficacious vaccine with the potential of homologous or heterologous boosting, and several HIV mRNA vaccine candidates have started clinical trials. The lecture ended with the emphasis that vaccine development at its inception should focus on access and equity alongside its efficacy—“Vaccination saves people, not the vaccine”!

Dr. **Paolo Palma** (Department of Pediatrics, Children Hospital/Ospedale-Pediatrico-Bambino-Gesu, Italy) gave an important talk on the perspectives of therapeutic vaccination against HIV-1 for perinatally HIV-1 infected patients. The talk outlined the need for and the importance of therapeutic HIV vaccines, the aim of such vaccines, and the ongoing work that has been done in this field. Nearly 2 million children aged 0 to 14 worldwide live with HIV-1 (UNAIDS 2023 https://www.unaids.org/en/resources/fact-sheet, accessed on 27 June 2023; [15]). Their well-being largely depends on their access to antiretroviral therapy, which they will have to continue for the entirety of their lives. Even though antiretroviral therapy suppresses HIV viremia, these medications have both short and long-term toxicities, and there is poor adherence or even a lack of access to the therapy. This, together with the fact that cases of spontaneous HIV remission are very rare, even in early-treated children, highlights the need for a therapeutic vaccine. Dr. Palma further focused his talk on the HIV Vaccine to Reduce Reservoir in Children & Adolescents (HVRRICANE) study by the EPIICAL Network. This study is currently recruiting early-treated children due to their low viral reservoir and low viral diversity. Participants receiving antiretroviral therapy will receive an HIV DNA vaccine (HIVIS-DNA) and a multigenic, recombinant modified vaccinia Ankara-HIV-1 vaccine candidate (MVA-CMDR). In addition, a subgroup will receive the approved HPV vaccine Cervarix, which contains a toll-like receptor-4 agonist, which is expected to serve as an adjuvant to boost the immune responses to the HIV vaccine. This study will evaluate specific T- and B-cell responses, perform immunophenotyping, transcriptomic and proteomic analysis, and analyze antibody-defendant cellular cytotoxicity of the participants in the study.

Dr. **Frank Msafiri** (Muhimbili University of Health and Allied Sciences, Dar es Salaam, Tanzania) presented a review of HIV-1 vaccine trials in Sub-Saharan Africa, where 51% of new HIV infections are registered, to highlight the efforts made in the search for interventions which could help achieve sustained control of the epidemic in this region. The review revealed that only 12% of all vaccine trials had been conducted in the region, indicating the need for more phase 1 and 2 clinical trials to reinforce the local scientific capacity to achieve this goal.

The session’s final talk was given by Dr. **Massimiliano Bissa** (National Cancer Institute, Bethesda, Rockville, MD, USA), who examined the immune correlates of SIV-acquisition risk in macaques vaccinated with a promising HIV vaccine candidate. The macaque-SIV model has been extensively used in HIV research and has proven relevant as it recapitulates several correlates of the risk of HIV-1 infection identified in the human HIV recombinant canarypox–derived vector (ALVAC) vaccine trials. These include antibodies against SIV/HIV variable region 2 (V2), CD4+ cells, and antibody-dependent cellular cytotoxicity (ADCC). The macaque-SIV model has been further improved using DNA encoding an SIV envelope protein gp120 with deleted V1. V1 and V2 are amino acid loops found at the apex of the HIV gp120 trimer. Previous research demonstrated that V1 antibodies are associated with an increased risk of SIV infection in macaques [16]. Since CD14+ monocytes have been repeatedly demonstrated to be a correlate of reduced risk of infection in the recipients of the DNA/ALVAC/gp120/alum vaccine, the current study aimed to elucidate whether the vaccine efficacy could be linked to the epigenetic reprogramming of CD14+ monocytes. Previous studies by the group indicated that responses mediated by activation of RAS, guanosine triphosphatases that regulate cell growth, differentiation, and apoptosis, correlated with reduced virus acquisition. Based on this, the current study also aimed to elucidate whether insulin-like growth factor 1 (IGF-1), a RAS activator, could affect the vaccine-evoked responses. These hypotheses were tested by canonical immunoassays (flow cytometry, Luminex assays, etc.), transcriptome analyses, and chromatin accessibility evaluation in the macaques immunized with the DNA/ALVAC/gp120/alum vaccine, with or without IGF-1. The results reconfirmed the role of ADCC and V2-specific ADCC specifically in decreasing the infection acquisition risk.

Additionally, the study identified that efferocytosis correlates with a reduced risk of infection. Efferocytosis is a CD14+ characteristic process of clearing engulfed apoptotic cells. Vaccination was shown to change the chromatin accessibility of CD14+ cells. Changes to the accessibility of cAMP response element-binding protein 1 (CREB1) were linked to the level of V2-specific ADCC. The co-administration of the vaccine with IGF-1 resulted in the increase of V2-specific antibodies and the frequency of CD14+ cells, although a significant boost in vaccine efficacy could not be observed. Overall, the study reported by Bissa et al. demonstrated that epigenetic reprogramming of CD14+ cells and its effect on efferocytosis contribute to vaccine efficacy by effectively removing infected apoptotic cells. Additionally, IGF-1 seems to synergize with the ALVAC-SIV/gp120/alum vaccine, but not sufficiently enough to significantly improve vaccine efficacy. The presented results will be integrated into the phase 1 HIV Vaccine trial CLEAR (Combined Long-term Efferocytosis and ADCC Responses).

## 4. Session 3: HPV Infection & HPV Vaccination

This session covered the impact and attitude toward HPV vaccination in the general public and people living with HIV-1, HPV prevention strategies, and HPV markers. The session was chaired by Prof. Gunta Lazdane (Riga Stradins University, Latvia).

The session was opened by a plenary lecture about the population-level impact that the introduction of HPV vaccination has had given by Prof. **Tina Dalianis** (Karolinska Institutet, Solna, Sweden). Prof. Dalianis is a renowned researcher and clinician in the field of tumor virology who strongly promoted the introduction of prophylactic HPV vaccines in Sweden. The lecture overviewed HPV prevalence studies done in Sweden before and after the introduction of HPV vaccination. Firstly, she presented a study on the prevalence of HPV and oropharyngeal squamous cell carcinomas in Sweden. HPV was not considered a risk factor for head and neck cancer development until 2000. Retrospective studies on pre-treatment biopsies of tonsillar and base tongue cancer (two major oropharyngeal cancers) revealed an increase in the proportion of HPV-positive cancers alongside the rise in cancer cases. Among tonsillar cancer cases, the percentage of HPV-positive cancers increased from 23% to 93% (2000–2007), and for the base of tongue cancers from 58 to 84% (1998–2007) [17,18]. Analysis revealed that for other head and neck cancers, the association with HPV was not that significant [19].

Interestingly, tonsillar and base tongue cancer patients with HPV-positive tumors were shown to have better disease-specific survival compared to those with HPV-negative tumors [19,20,21]. These findings indicated the importance of vaccination not only in girls but also in boys. The virus is a major cause of cancer in the head and neck region, anus, vulva, vagina and penis. Thus, men are also at risk. Furthermore, the incidence of oro-, naso-, hypo-pharynx cancer and cancer of the lips and oral cavity in man are several fold higher than in women [22]. The survival data highlighted the importance of individualized treatment and not overtreating patients with a better prognosis and a disease that is different in many ways. The second major study presented by Prof. Dalianis involved a survey of HPV prevalence in the genital tract and oral samples of adolescents attending the youth clinic at the beginning of the HPV vaccination campaign in Sweden (2007–2011). Cervical samples demonstrated a high prevalence of HPV, HPV 16 in particular (35% of the tested samples). 9.5% of oral samples harbored HPV, most of which were high-risk HPV-positive, and 17% of the women with genital HPV had oral HPV infection with a concordant HPV genotype [23,24]. Similar studies were conducted during 2013–2015 and 2017–2018 [25,26]. These later studies vividly confirmed the impact of HPV vaccination, as the prevalence of the cervical HPV16 infection dropped considerably, 5% in vaccinated women, as opposed to the previously reported 35%. The prevalence of HPV in the oral samples has also dropped from 9.5 to 1.5%. The latest data shows a further considerable drop in HPV 16 and HPV 18 prevalence among vaccinated and nonvaccinated individuals. However, infections with the high-risk HPV 39, 51, 56, and 59, not included in the nonavalent Gardasil-9 vaccine (L1 of HPV 6, 11, 16, 18, 31, 33, 45, 52 and 58), remain and may present a risk of HPV-related cancers despite the introduction of the nonavalent HPV vaccine.

Dr. **Zane Kivite-Urtane** (Riga Stradins University, Riga, Latvia) presented a study on the knowledge of HPV vaccines and the attitude toward HPV vaccination among women in Latvia. Organized HPV vaccination was introduced in Latvia in 2010, and its coverage has reached 69%, yet the incidence and mortality rates of cervical cancer remain relatively high in Latvia. This study by Dr. Kivite-Urtane et al. aimed to evaluate the awareness and attitude toward HPV vaccination and its associated factors. Among 1313 women aged 25–70, 71% have heard of the vaccine against HPV, 52% considered it necessary to vaccinate young girls, and only 3% reported that they had been vaccinated. Multivariate analysis revealed that the lack of awareness was associated with lower education, lower income, and women with no chronic health conditions or STIs. A negative attitude toward vaccination was similarly associated with a lower income and with never or rarely visiting a gynecologist, showing for this group of subjects the lack of a secondary prevention strategy (a pap test-based cervical screening) and a primary prevention strategy (HPV vaccine). These results illustrate the need for a more far-reaching public awareness campaign that would inform of the risks of HPV and for the ways to prevent it not only by the centralized actions of the public health agencies but by involving local medical personnel, such as general practitioners, whom people are seeing frequently and with whom they can discuss all aspects of vaccination.

The next talk by Prof. **Anna-Lise Williamson** (University of Cape Town, Cape Town, South Africa) covered the importance of HPV screening and vaccination in general and in more depth in people living with HIV-1. The talk highlighted the burden of HPV-associated cancers worldwide, HIV-1 infection in sub-Saharan Africa, especially in girls and women, the current vaccination approaches, and the important interplay between HIV-1 and HPV. Earlier studies have found that HIV-positive men and women are more frequently HPV-positive than HIV-negative individuals and often harbor multiple HPV types. HPV infection has been proposed to be one of the risk factors for HIV-1 acquisition; HPV-infected individuals have two times higher risk of HIV-1 infection than HPV negative. Symmetrically, HIV-positive women have a higher prevalence of persistent HPV infection than HIV-negative; the latter increases the risk of developing pre-caner conditions and cancer. HPV vaccination in people living with HIV-1 has demonstrated that these people develop anti-HPV antibodies at low levels. The clinical significance of these findings is unknown, and more research is needed to evaluate the effect of HPV antibodies on HPV infection and the progression of chronic HPV infection in HPV-infected HIV-positive individuals. The presentation was concluded by the preliminary group results obtained in a study analyzing HPV genotypes in adolescents and young adults living with HIV-1 who have received the Gardasil quadrivalent HPV vaccination. The preliminary analysis of the genital swabs of 103 participants revealed that only 32% of them were HPV DNA negative. HPV genotypes included in the Gardasil vaccine could be detected in 17, with HPV 18 in 14 of 103 participants. These results indicate the need to continuously monitor high-risk HPV types in people with HIV-1 to mitigate future disease development.

Continuing on the topic of HPV and people living with HIV, Dr. **Sona Chowdhury** (University of California, San Francisco, CA, USA) reported the results of a study analyzing HPV genotype distribution in people living with HIV as determined by anal screening. The study aimed to evaluate the prevalence of specific high and low-oncogenic risk HPV types in anal cancer and anal precancerous lesion (high-grade anal squamous intraepithelial lesion, HSIL) cases in people living with HIV-1 compared to HIV-negative individuals. The study included HIV-positive and negative controls with benign anal epithelium. Results demonstrated that HPV 16 is the main etiological factor of precancerous lesions and anal cancer development, irrespective of HIV serostatus. Yet, other non-HPV16 oncogenic HPVs (18, 31, 51) also play a role in the disease. Multiple HPV-type infection was a hallmark feature in all tissues of people living with HIV-1. Benign tissues from people living with HIV-1 had a higher proportion of both high and low-risk HPV types, indicating that HIV status influences HPV infection and prevents HPV clearance, thus increasing the likelihood of developing anal cancerous lesions.

The session was closed with a presentation by one of the early career researchers, Dr. **Andrea Cerasuolo** (Istituto Nazionale Tumori, Naples, Italy), who reported the results of the study on the role of E6* isoform and the levels of their expression as markers of active HPV infection in the head and neck cancers. The oncogenicity of HPV is largely dependent on two of its proteins, E6 and E7. E6 and E7 oncogene mRNA is subjected to alternative splicing, resulting in different isoforms. The most abundant isoforms are E6*I and E6*II. The latter was shown to be expressed at different stages of cancer development, to be differentially regulated, and to produce proteins with varying capabilities to interact with other viral and cellular proteins [27,28]. HPV-positive head and neck cancers are distinct from their HPV-negative counterparts in their molecular and clinical characteristics, such as the decreased mutation rate and the better response to treatment and overall prognosis [29]. HPV DNA positivity alone cannot distinguish whether head and neck squamous cell carcinomas (HNSCC) are related to the infection; it needs to be paired with additional molecular markers. Knowing this, the reported study by Cerasuolo et al. aimed to distinguish HPV-related and unrelated HNSCCs by such parameters as the presence and load of HPV DNA, expression of HPV16 oncogene E6 and E6*I isoforms, and the correlations between the two. HPV16 was found to be the most frequently detected genotype in HNSCCs. E6*I isoform was detected in 50% of the investigated HNSCCs, including all HPV-positive oropharyngeal and nasopharyngeal SCCs. The expression level of HPV16 E6*I correlated with HPV DNA positivity in HNSCCs with medium to high viral loads. The study concluded that evaluating various HPV markers—load, and oncogene isoform expression, is the most efficient approach to identifying HPV-related HNSCCs and could serve as a basis to personalize cancer treatment for these cases.

## 5. Session 4: Cancer Vaccines

The fourth session of the conference addressed the molecular basis of cancer vaccine development, the advances in developing HPV therapeutic vaccines, and vaccines against liver cancer. The session was chaired by Prof. Franco M. Buonaguro.

The session was opened by Prof. **Joel Palefsky** (University of California, San Francisco School of Medicine, USA), an internationally recognized expert in the biology, treatment, progression, and history of HPV infections and associated cancers. His plenary lecture on the therapeutic vaccines for treating chronic HPV Infection and associated cancer covered the basis of HPV-related precancer and cancer, the challenges in developing therapeutic vaccines, and the new concepts evolving in this field. The life cycle of HPV is linked to the differentiation status of the cell it infects. HPV infects basal cells of the stratified epithelium through micro-abrasions. Viral gene expression is tightly regulated as the infected basal cell migrates toward the epithelial surface, starting from the early genes which facilitate viral replication and cell re-entering into the S phase and ending with the expression of late genes that encode structural proteins. The HPV-caused transformation of the epithelium can be classified using a two-tiered system—low-grade and high-grade squamous intraepithelial lesions (LSIL/HSIL), which are followed by the development of carcinomas. Since at least 5% of all cancers can be attributed to HPV infection [30], the development of a therapeutic HPV vaccine is crucial. However, it faces several challenges. Firstly, the site of the infection is a sequestered compartment. Secondly, viral proteins are expressed at a low level and have poor immunogenicity. Thirdly, the virus is prone to immune escape. Last but not least, are difficulties in defining which stage of the disease should be targeted - cancer or precancer as well as difficulties in recruiting participants with HSIL. Prof. Palefsky highlighted the importance of identifying the precancer (HSIL) cases to prevent cancer, as was revealed in the results of the ANCHOR trial. This trial demonstrated that timely diagnosis and treatment of anal HSIL had prevented the development of anal cancer: the median follow-up of 25.8 months revealed a 57% reduction in the anal cancer cases [31]. The second part of the lecture covered the progress made in developing vaccines against HPV-associated cancer, various vaccine delivery methods, and immunogens used, with an in-depth dissection of the best vaccine candidates. Many approaches have been explored to develop therapeutic vaccines against head and neck cancer [32]. Most vaccine candidates against HPV-related HNSCCs or HPV-related cancers, in general, target the oncoproteins E6 and E7, which are consistently produced by the cancer cells and contain immunogenic epitopes. Some have also used p16, an overexpressed marker in HPV-related HNSCCs. Several vaccine delivery methods—peptide, nucleic acid, and cell-based are currently in trials for HNSCCs. Similarly, E6 and E7 are the most often used antigens in developing therapeutic vaccines against cervical intraepithelial neoplasia (CIN). A recent meta-analysis reviewed therapeutic cervical cancer vaccines targeting the early stages of the disease (CIN) [33]. The reviewed studies revealed that therapeutic vaccines could induce immune responses, leading to the regression of CINs and the elimination of HPV DNA. The lecture specifically covered three of the HPV-associated cancer therapeutic vaccines. First, VGX-3100—a vaccine containing plasmid DNA encoding the E6 and E7 proteins of HPV16 and 18. The vaccine is administered intramuscularly via electroporation. Phase 2 trials for CIN2/3 demonstrated that 48.2% of 114 VGX-3100 recipients had histopathological regression of the lesions, and for those who had regressed after six months, tests showed no HSIL recurrence at 18 months. Additionally, most of those who had regressed at six months had no detectable HPV16 or 18 at 18 months after the completion of treatment [34,35]. Phase 3 trial of the vaccine reported a significantly higher percentage of HSIL regression and HPV clearance at 36 weeks in the treatment vs placebo group (23.7% of 131 vs. 11.3 % of 62). The second, ISA101, is a peptide vaccine composed of 12 synthetic peptides derived from the E6 and E7 oncogenic proteins of HPV 16, which are up-taken and presented by the dendritic cells. A study analyzing the effects of this vaccine on vulvar intraepithelial neoplasia found that after 12 months, 15 of 19 patients had clinical responses, and a complete response was observed in 9 of 19 patients. All patients had vaccine-induced T-cell responses. Patients with a complete response at three months had a significantly stronger interferon-γ–associated proliferative CD4+ and CD8+ T-cell response than non-responders [36]. In the studies where ISA101 was combined with the checkpoint inhibitor nivolumab, a response rate of 33% was demonstrated in patients with incurable HPV-16+ cancer. In the long-term follow-up study, the response had a median duration of 11.2 months, and 38% of patients with detectable responses demonstrated no disease progression after three years [37,38]. The third vaccine, VTP-200, is a multi-genotype therapeutic vaccine, the clinical trials of which are still ongoing. VTP-200 carries E6, E7, E1, E2, E4, and E5 gene segments of HPV16, 18, 31, 35, 52, and 58 via two non-replicating viral vectors—chimpanzee adenovirus Oxford 1 and modified vaccinia virus Ankara. To conclude, Prof Palefsky discussed other immunotherapeutic strategies for HPV-related diseases, like T-cell receptor (TCR)-engineered T-cells. TCR-engineered T-cells targeting E7 have demonstrated HPV-positive tumor cell killing in vitro and clinical trials. In a phase 1 trial, the TCR-engineered T-cell treatment showed robust tumor regression in patients with metastatic HPV-associated cancers [39].

The following talk on the development of the vaccine against hepatocellular carcinoma (HCC) was given by Prof. **Luigi Buonaguro** (National Cancer Institute ‘Fondazione Pascale’, Naples, Italy). He reviewed the existing immunotherapies for HCC, future perspectives in this field, and his group’s extensive experience in HCC vaccine development and HCC-related research. The need for new treatment options was highlighted by the fact that liver cancer is the second most common cause of death from cancer, and the available treatment efficacy is limited, especially in the later stages of the disease. The development of HCC therapeutic vaccines is hampered by several limiting factors. Firstly, the tumor microenvironment of HCC is quite immunosuppressive. T regulatory cell infiltration, the Kupffer cell produced anti-inflammatory molecules (TGF-β, IL-10), several cell-expressed inhibitory molecules like PD-L1, and other mechanisms which impair adaptive immune responses [40]. Secondly, the antigens that could be used in the vaccines are few—tumor-associated antigens used in the previous clinical trials, are limited in number, and are not specific to liver cancer.

Furthermore, tumor-specific antigen-targeting vaccines need to be tailored to individual patients. These challenges are illustrated by the fact that most HCC vaccine trials have stopped at Phase 1/2, as they could not demonstrate sufficient immunogenicity to move to later clinical phases [41]. Prof. Buonaguro further discussed the background and results of the HepaVac project [42]. HepaVac aimed to develop a new therapeutic peptide vaccine based on the newly discovered HCC tumor-associated-antigens, thus improving clinical outcomes for HCC patients after standard treatment. Tumor-associated-antigens were identified by RNA and HLA ligand (common HLA haplotypes were chosen—HLA-A*02 and A*24) analysis in tumor samples and adjacent normal tissues, epitopes expressed only in tumor samples were chosen and tested for immunogenicity using artificial antigen-presenting cells and PBMCs from patients and healthy donors. Twelve HLA class I peptides and 4 HLA class II peptides were chosen for the vaccine formulation and were adjuvanted with a novel RNA-based adjuvant. In the phase 1/2 multicenter trial, patients with early primary liver cancer, with HLA-A*02 and A*24, after standard treatment, received the vaccine and were monitored for up to three years.

Twenty patients completed the study, among whom only 2 had disease recurrence. Overall, the vaccine showed a good safety profile and induced immune responses against both class I and class II peptides [43]. Future development of the vaccine will aim to increase its efficacy by formulating it with a different adjuvant (montanide) and combining it with an immune checkpoint inhibitor (durvalumab). In parallel, L. Buonaguro and his group aim to improve the immunogenicity and antigenicity of tumor-associated antigens through several research approaches. One approach foresees the generation of heteroclitic peptides—epitope-containing peptides with one or two amino acid residue substitutions in the epitope sequence to increase their immunogenicity. A study exploring this approach with Trp2 and HPV E7 peptides demonstrated stronger binding of heteroclitic peptides to MHC molecules than parental peptides [44].

Additionally, immunization of mice with such peptides induced an immune response that significantly delayed tumor growth compared to the wild-type peptides. The other approach foresees the identification of non-self-antigens that can elicit cross-reactive T-cell responses against tumor antigens—molecular mimicry. This approach was explored in a recent study where HCC-specific tumor-associated antigens were shown to have high homology with viral antigens. The epitopes identified in this study were highly homologous in the amino-acid sequence level and 3D conformation. The study also demonstrated that T-cells cross-reacted to the paired epitopes [45]. Overall, these homologous viral antigens present new targets for immunotherapy as they are unaffected by central immune tolerance and elicit potent T-cell responses. Based on these promising results, a patent has been filed, and a vaccine encompassing these viral antigens is in development and planned to be tested by the end of 2023.

Prof. **Margaret Liu** (Karolinska Institutet, Stockholm, Sweden & University of California, San Francisco, USA) lectured on the DNA and mRNA vaccines for cancer, the rationale behind their design, their mechanism(s) of action, and progress in their development. Prof. Liu described the types of immune responses effective against cancer, the rationale of vaccines and immunotherapy, characteristics of DNA and mRNA vaccines that may make them helpful in fighting cancer, and ongoing clinical trials with examples of types of cancer that a vaccine can target. Cancer vaccines exploit the ability of the immune system to recognize and destroy neoplastic cells through the recognition of either viral antigens (in the case of infection-induced cancers, such as HPV E6 and E7) or antigens that are overexpressed by the transformed cells. Today, the development of cancer vaccines is supported by the success of immune interventions such as specific antibodies, immunostimulatory molecules, CAR T-cells, and immune checkpoint inhibitors. Next, Prof. Liu discussed the development prospects of DNA and mRNA-based cancer vaccines. The highlighted advantages of using nucleic acid cancer vaccines involve generating adaptive immunity while stimulating innate immune responses, the rapid construction and manufacturing, their capacity to deliver multiple antigens and co-deliver cytokines, and the possibility to design patient-specific vaccines. Prof. Liu listed multiple nucleic acid cancer vaccines, based on both mRNA and DNA, currently in trials and concluded her talk with future directions of immunotherapeutic vaccines. Prospects include the co-administration of immunostimulatory molecules, the expansion of patient-specific vaccine constructs, and the exploration of heterologous prime-boost vaccinations (DNA/mRNA, DNA/protein, etc.).

In the next talk, Prof. **Joon Haeng Rhee** (Chonnam National University, Gwangju, Republic of Korea) returned to therapeutic vaccines against HPV by reporting on an all-in-one adjuvanted therapeutic cancer vaccine approach targeting antigen-presenting cells to induce potent antitumor cellular immune responses. Prof. Rhee reported the results of developing a therapeutic cancer vaccine targeting antigen-presenting cells consisting of a dendritic cell targeting peptide, the antigen, and an adjuvant [46]. The study screened a phage display library and found a novel peptide (DCpep6) that specifically binds to and is internalized by dendritic (CD11c+) cells. That was followed by the engineering and testing a therapeutic cancer vaccine consisting of the identified peptide (DCpep6), an optimized HPV E7 tumor antigen, and a previously extensively studied flagellin adjuvant, all contained in a single molecule. Results reported by Rhee et al. demonstrated that the all-in-one vaccine was stably expressed, with each of the components retaining their functionality, and was successfully taken up by the dendritic cells. Similarly, in vivo, experiments demonstrated the rapid biodistribution of the vaccine into draining lymph nodes and internalization into the dendritic cells. Furthermore, immunization with the proposed vaccine induced robust tumor-killing T-cell responses, tumor growth inhibition, and increased long-term survival of mice implanted with syngenic TC-1 tumor cells.

One of the early career researchers Ms. **Alesja Dudorova** (Pauls Stradiņs Clinical University Hospital & Riga Stradins University, Riga, Latvia), presented a study exploring murine adenocarcinoma cells expressing HPV 16 oncoproteins as a potential model of HPV-associated carcinogenesis. In the study, she produced subclones of murine mammary gland adenocarcinoma 4T1 cells expressing HPV16 E6/E7 and studied their properties in vitro and in vivo by implanting them into BALB/c mice. In vitro experiments revealed that the expression of transcription factors—HIF-1α, Twist, and Nrf2 was modulated by the expression of E6 and E7. Oncogene expression promoted cell passage of the G2/M cell cycle checkpoint. The expression of HPV oncoproteins also caused double-stranded (genomic) DNA damage. Overall, the in vitro observations, such as DNA damage, accelerated cell cycle passage through the G2/M checkpoint, and changes in the expression of several transcription factors characteristic to the epithelial–mesenchymal transition, coincided with the ones observed in HPV-associated squamous cell carcinomas. In vivo, experiments in mice showed the ability of E6/E7 expressing 4T1 cells to form solid adenocarcinomas and infiltrate into the liver and lungs. E6/E7 expressing 4T1 cells did not differ from the parental cells in the number or size of induced liver metastasis. The metastatic and tumorigenic activity of the E6/E7 expressing cells correlated with the decrease in the cell cycle’s proportion of cells in the G2/M phase. Overall, the 4T1 subclones expressing HPV16 oncoproteins exhibited several critical features of HPV-associated cancers. Due to this, such cells can be useful in the tests of candidate vaccines against chronic HPV infection and HPV-associated cancer.

The session was concluded with another study on HPV therapeutic vaccine research given by Dr. **Juris Jansons** (Latvian Biomedical Research and Study Center, and Riga Stradins University, Riga, Latvia), who presented results of DNA immunization of mice with HPV 16 E6 and E7. The study by Jansons et al. evaluated the immunogenicity of the consensus HPV16 E6 and E7 designed after the analysis of HPV16 E6/7 sequences isolated from 35 HIV, TB or HIV/TB coinfected patients, which were found to be identical to the sequence of the reference HPV16 strain except for single amino acid substitutions [47]. Namely, 50% of the analyzed sequences of E6 contained substitutions of either R17G or L90V, or both. E6 and E7 sequences were cloned into pVax1 and used for DNA immunization of C57bl6 mice by a prime/boost regimen (intradermal injections at days 1 and 21 followed by electroporation), cellular immune responses were assessed ten days post boost. The CD4+ T-cell responses were observed against a cluster of E6 epitopes of HPV 16, while an anti-HPV16 E7 immune response was not detected.

Interestingly, only the E6 peptide representing the R17-variant was recognized by CD4 T-cells, while the G17-variant was not, indicating that this substitution may represent an immune escape mutation. Next, the study assessed whether DNA immunization of mice with E6/E7 confers protection against E6/E7-expressing tumor cells syngenic to the immunized mice. For this, BALB/c mice were immunized as described above and, 11 days after the boost, were challenged with subcutaneous injections of murine adenocarcinoma 4T1luc2 cells made to express oncoproteins E6 and E7 of HPV16. A reduction in the tumor size and weight and the number of liver metastasis was observed in the BALB/c mice immunized with HPV16 E6 but not with HPV16 E7. The results of the reported study are of high importance in future HPV therapeutic vaccine development.

## 6. Session 5: Facing New and Re-Emerging Threats

The fifth session was devoted to the development of vaccines and vaccine-related research involving infectious agents such as the Crimean-Congo Hemorrhagic Fever Virus, hepatitis B virus, influenza virus, and tick-borne encephalitis virus, with new challenges and new approaches. The session was chaired by Prof. Anke L. W. Huckriede.

The session was opened by Prof. **Felicity Burt** (University of the Free State, Bloemfontein, South Africa), an expert in the field of host immune responses to arboviral and zoonotic infections, with the presentation on the advances in the field of Crimean-Congo Hemorrhagic Fever Virus (CCHFV) vaccine development. Climate change has strongly influenced the spread of tick-borne and mosquito-borne diseases [48]. The increase in temperature and decrease in the precipitation and humidity have also resulted in an increased incidence of CCHFV infections, further necessitating the introduction of CCHFV vaccination [49,50]. The lecture by Prof. Burt covered the discovery, epidemiology, and biology of CCHFV, the need for vaccination, and advances in the field. CCHFV is a tick-borne infection prevalent in Africa, Asia, Europe, and the Middle East. Infection can be acquired through tick bites or after contact with infected blood or tissues of either patients or livestock. The disease has a fatality rate of 24%. The distribution of the virus correlates with the distribution of its vector, the ticks of the genus *Hyalomma*. In recent years, the distribution of these ticks has expanded. Since there is no approved vaccine or antiviral treatment, CCHFV is listed among WHO priority pathogens [51]. In recent years, more studies have explored novel vaccines due to the development of new mouse models susceptible to the infection and exhibiting pathology similar to that observed in humans; non-human primate models have also been explored. Several vaccine types have been tested with varying efficacy and antibody response—sub-unit, transgenic plant, virus-like particle, nucleic acid, inactivated virus, and recombinant viral vector vaccines. The vaccines tested so far have exploited various CCHFV antigens—glycoproteins, nucleoproteins, and viral polymerase. Virus-like particle vaccines expressing either viral glycoproteins, nucleoproteins, or polymerase proteins, rendered protection against viral challenges with survival rates ranging from 40% to 100%. DNA-based vaccines expressing glycoprotein precursors or nucleoprotein ensured survival rates of 50% to 100%, which was recently demonstrated by Hawman et al. [52,53] and viral vector (MVA or recombinant adenovirus) vaccines, survival rates up to 100 % for the vaccine based on viral glycoproteins, and no protection for vaccines based on viral nucleoproteins. Summarizing the results of the trials of several vaccine candidates, the presence of neutralizing antibodies does not necessarily correlate with protection, and the protection most probably requires both B- and T-cell responses. Overall, more work needs to be done to elucidate the immune correlates of protection. Epidemiological data exploring the infection pathways has brought attention to certain groups, such as people working closely with large animals (in farms, slaughterhouses, and veterinary clinics) [54] as well as hospital workers, being at high risk of CCHFV infection, and highlighted the fact that vaccination of targeted population may be the best approach to mitigate the CCHFV outbreaks.

The next talk by Prof. **Anke L. W. Huckriede** (University Medical Center Groningen, Groningen, The Netherlands) highlighted the utilization of in vitro systems in vaccine quality assessment and elucidating vaccine-induced immune responses. Prof. Huckriede began the lecture by stating that the use of animals is crucial for vaccine development research. However, due to the inherent differences between human and animal models (mice, non-human primates, etc.), the latter are often poor predictors of vaccine-induced immune response and vaccine efficacy in humans. With this in mind, Prof. Huckriede discussed in vitro methods that can be used in research to model the responses of the human immune system. In vitro systems for immune response modeling range from simple cell cultures to whole tissues or organoids. The presentation focused on the use PBMC based in vitro cultures. Such blood-cell systems use PBMCs isolated from blood, which then can be sorted into specific subpopulations and cultivated or cultivated in co-cultures. Basic readouts of such systems include biomarker expression, cytokine detection, and transcriptomic analyses to determine the activation of certain cells and specific molecular pathways. The talk focused on two examples where these PBMC-based systems have been used.

In the first example, the PBMCs were used to compare two types of influenza vaccines—whole inactivated virus and subunit vaccine. Monocyte-derived dendritic cells and T-cells were used to evaluate these two vaccines. The results coincide with in vivo experiment observations—the inactivated virus vaccine was more immunogenic. The whole inactivated virus, not subunits, activated immune-related gene expression pathways in the dendritic cells and induced higher cytokine secretion [55]. Both vaccines could stimulate CD4 and CD8 T-cell responses but to different extents. The vaccines activated T follicular helper cells, essential in humoral immunity establishment [56]. In the second example, PBMCs were used to assess the quality of the inactivated tick-borne encephalitis virus (TBEV) vaccine batches. Human PBMCs responded to the TBEV vaccine by upregulating innate immune responses and antiviral defense (interferon) signaling. The observed immune responses were consistent among various vaccine batches, and the in vitro system based on PBMC cultures could detect differences in batches with high resolution [57,58]. The presentation concluded that human PBMC-based systems are promising tools in developing vaccines, including assessment of immune response, bypassing the use of animal models.

Dr. **Charlotta Nilsson** (The Public Health Agency of Sweden & Karolinska Institutet, Sweden) returned to the topic of HIV-1 and presented a study exploring the effects of HIV vaccine-induced antibody responses on the accuracy of HIV testing algorithms. HIV diagnosis in many resource-restricted countries depends on rapid diagnostic tests—tests detecting various HIV-1 antigen-specific antibodies. Many HIV-1 vaccine candidate trials take place in Sub-Saharan Africa. The study by Nilsson et al. aimed to explore the impact of vaccine-induced seroreactivity on the performance of HIV-1 rapid diagnostic tests via two HIV diagnostic algorithms in use in African countries. The study used serum samples from the participants of phase 1/2a HIV vaccine trials evaluating a prime-boost DNA-modified vaccinia virus Ankara-Env protein vaccine strategy. One hundred thirty-seven serum samples were collected at the peak of immunogenicity—1 month after vaccination, and some samples were also collected over time—1 month, 16 months, and three years after 3 × HIV-DNA + 2 × HIV-MVA vaccination. The study exploited testing algorithms used in Tanzania (SD Bioline HIV1/2 for screening and Uni-GoldTM HIV-1/2 for confirmation) and Mozambique (Alere DetermineTM HIV-1/2 for screening and Uni-GoldTM HIV-1/2 for confirmation). The results revealed that the Tanzanian diagnostic algorithm would have misdiagnosed as HIV-1 infected 54%, and the Mozambican—26% of the vaccinated healthy volunteers. However, the number of misdiagnoses decreased over time, and three years after vaccination, none of the 20 serum samples collected at that time tested HIV-1 positive using rapid diagnostic tests [59]. The reported results suggested that the HIV diagnostic algorithms used in these countries would have misclassified the healthy vaccine recipients as HIV-positive. Thus, it is necessary to develop affordable testing tools to differentiate vaccine-induced seroreactivity from cases of HIV-1 infection.

Dr. **Tatjana Tallo** (The Public Health Agency of Sweden, Solna, Sweden) addressed the public need for the Hepatitis B virus (HBV) vaccine. Her talk overviewed the importance, challenges, and future perspective of vaccination against HBV. HBV remains a major public health problem, estimated to cause 30% of cirrhosis and 53% of liver cancer cases. The first recombinant subunit vaccines for HBV were licensed at the end of the 1980s, and in 1992 WHO recommended including these vaccines in all national vaccination programs. Vaccination in infancy is critical since chronic infection risk is age-associated—nearly all infected early in life develop a chronic infection as opposed to only 5% of those infected in adulthood. The introduction of HBV vaccination has had a significant impact—between 1990 and 2019. The all-age chronic HBV prevalence declined by approximately 31% and among children five years and younger by 77% [60]. WHO has outlined four strategies for HBV immunization: (1) Universal vaccination for infants within 24 h of birth; (2) Full immunization of infants by routine vaccination; (3) Catch-up immunization of unimmunized groups (4) Monitoring of vaccination progress and impact. Even though vaccination has been introduced for a while and demonstrated significant effects in most countries, several challenges are associated with these strategies. First, vaccination within the first 24 h of birth is hard to achieve in low-resource countries due to the high rates of births outside healthcare facilities, vaccine storage requirements, and low awareness of parents about vaccination benefits and importance. Health promotion campaigns and the development of heat and freeze-stable HBV vaccines are thus crucial in these regions. Also, full immunization may not be achievable since certain parts of the population lack response to vaccination, like elders and immunocompromised individuals, highlighting the need for HBV vaccine modification with various antigens or adjuvants. Finally, the duration of protection after vaccination is not precisely known yet, and more long-term studies are necessary. The lecture was closed with the answer to the question posed in the title of the presentation—“Do We Need Hepatitis B Vaccine?”—“YES!”.

Continuing on HBV vaccination, Dr. **Irina Sominskaya** (Latvian Biomedical Research and Study Centre, Riga, Latvia) reported the study results of the combined therapeutic/prophylactic Hepatitis B VLP vaccine prototype. The developed and accepted HBV vaccines are HBV surface antigen (S) targeted, and even though their prophylactic activity is very high, they have no therapeutic activity. WHO estimates that almost 300 million people live with chronic HBV [61]. Their curation could be aided by therapeutic HBV vaccination. The development of a therapeutic HBV vaccine would therefore have a significant effect on global health. The reported study aimed to test a recombinant HBV core antigen-based virus-like particle (VLP) bearing surface-exposed HBV preS1 epitopes as a novel therapeutic HBV vaccine candidate. A library of HBc-preS1 VLPs consisting of various length HBc proteins (full length and C-terminus truncated) and the well-known virus-neutralizing epitope of the HBV S gene inserted at position 78 was created. VLPs were produced in *E. coli*, and the best producers were selected for further experiments. Packaging experiments with the VLP formed with preS1 (20–47) insertion of full-length HBc with model oligonucleotides showed promissing results. Furthermore, immunization of BALB/c mice demonstrated that several VLP formulations could induce strong anti-preS1 antibody responses. The results present the HBc-preS1 VLPs as potent therapeutic vaccine candidates due to their ability to induce immune responses and packaging ability, enabling their use as carriers for nucleic acid-based molecular adjuvants.

## 7. Session 6: Novel Vaccine Vehicles, Adjuvants and Carriers

The final session of the conference covered in detail the basis of vaccine adjuvants as well as novel adjuvants, new vaccine delivery systems, and new vaccine vectors. The session was chaired by Dr. Irina Sominskaya and Dr. Juris Jansons (Biomedical Research and Study Centers, Latvia).

Prof. **Dennis Christensen** from Croda Pharma (Frederikssund, Denmark), a leader in the field of vaccine adjuvants and lipid delivery systems for human and veterinary applications, opened the session with a plenary lecture on adjuvants as means to achieve the necessary immune responses in preventive and therapeutic vaccination. The lecture highlighted the aspects of modern vaccination—presentation of the suitable antigen in a sufficient amount, in the correct conformation, to the proper immune cell population, accompanied by the right co-stimuli, and for a sufficient amount of time. It also listed the “must have” constituents of modern vaccines: (1) an antigen delivery system that facilitates the delivery to the right cells; (2) the antigen—a molecule or a molecular structure that can interact with the immune system and create a response; (3) the immunomodulator—a compound that affects the immune system that determines the strength, duration, and type of immune response generated. The lecture then focused more closely on two aspects of modern vaccination—delivery systems and adjuvants—the areas of specialization for Croda Pharma. The discovery of adjuvant components potentiating the effects of vaccines began in the 1920s, and in 1930 one of the most well-known adjuvants was discovered—aluminum.

Further adjuvant discovery flourished only in the 1990s. The large time gap correlates with the discoveries made in the mechanisms driving the function of the immune system as a basis for adjuvant development. Many adjuvants target the innate immune systems’ ability to recognize pathogen-associated molecular patterns (PAMP) via pattern recognition receptors. One of the most common adjuvants is the lipopolysaccharide (LPS)—an outer membrane component of gram-negative bacteria that induces toll-like receptor (TLR) signaling. Prof. Christensen described an adjuvant based on LPS—Phosphorylated HexaAcyl Disaccharide (PHAD), a fully synthetic equivalent to bacterial-derived Monophosphoryl Lipid A (MPLA), which has been proven to be safe and tolerable in over 30 clinical trials. He also listed other Coda Pharma-developed immunomodulators, minerals based on liposomes and lipids, saponins, PAMPs, and other immunomodulators. Next to adjuvants, vaccine delivery systems are essential to ensure the presentation of the antigen to the right cell for a sufficient amount of time. Croda has developed several vaccine and adjuvant delivery systems. They are based on liposomes, lipid nanoparticles, and high-purity emulsions. Two Croda-developed adjuvant systems were discussed in more detail.

CAF^®^01 is a two-component liposomal suspension composed of dimethyldioctadecylammonium (DDA) and C-type lectin receptor agonist—trehalose dibehenate (TDB). This formulation has been shown to induce both humoral and cellular immune responses when combined with protein or peptide-based antigens and has been proven safe and tolerable. CAF^®^01 has been used in 6 finalized phase 1 clinical trial for tuberculosis, HIV, malaria, and chlamydia. Also, it is in preclinical development for influenza, polio, group A Strep, respiratory syncytial virus, and SARS-CoV-2. CAF^®^09b is a three-component liposomal suspension based on DDA combined with and C-type lectin receptor agonist—monomycoloyl glycerol (MMG) and TLR3 agonist poly I:C. This formulation has been shown to induce strong CD8+ T-cell responses when combined with protein or peptide antigens. It has proven safety and tolerability, and early studies indicated it could be a potential adjuvant for mucosal applications. CAF^®^09b has been used in 3 finalized phase 1 clinical trial for chlamydia, cancer neoepitopes, and B-cell lymphoma. CAF^®^09b used in vaccination against cancer neoepitopes and other tumor-associated antigens, has reached phase 2 clinical trials. CAF^®^09b is preclinical development for malaria, tuberculosis, influenza, and SARS-CoV-2. The lecture highlighted the necessity of effectively transferring knowledge and research results into biomedical products making vaccines more efficient.

Dr. **Anna Zajakina** (Latvian Biomedical Research and Study Centre, Riga, Latvia) gave a talk on the therapeutic modulation of tumor microenvironment with recombinant viral vectors overviewing how various viral vectors delivering immunomodulatory genes break the immunosuppressive tumor microenvironment (TME). Recently immunotherapeutic approaches such as immune checkpoint inhibitors and T-cell adoptive transfer have shown great promise in cancer therapy, yet patients who do not respond to such therapies remain. Unresponsiveness has been linked to the immunosuppressive TME—the tumor’s immune-responsive “cold” state. Such tumors are characterized by the lack of T-cell infiltration and oversaturation with immunosuppressive myeloid cells, among other characteristics. Thus, reprogramming the TME to an immune-responsive “hot” state could be vital to developing successful therapies. The use of viruses and viral vectors has been explored for this purpose in numerous preclinical and clinical trials [62]. There are several mechanisms behind virus antitumor activity—direct tumor destruction by oncolytic viruses, antiviral immunity induction of innate and adaptive immune responses against the tumor, and viral vector-encoded immune signaling molecule-driven TME reprogramming. The talk then focused more on the last mechanism—TME reprogramming, since studies have shown that oncolytic activity on its own is not enough to achieve stable therapeutic responses, and it needs to be supplemented with the delivery of TME reprogramming promoting therapeutic genes. Such therapies can either target the tumor immune cells or the tumor stroma. Reprogramming of tumor stroma involves the modulation of angiogenesis, the extracellular matrix, or the cancer-associated fibroblasts. Reprogramming of tumor immune cells can be achieved by delivering cytokines or other immune factors via viral vectors to increase M1/M2 macrophage ratio, tumor-antigen presentation by dendritic cells, T regulatory cell inhibition, cytotoxic T-cell/T regulatory cell ratio increase or otherwise stimulate the antitumor immune responses. Adenovirus, Poxvirus, Rhabdovirus, and other viral vectors containing such immune factors as TNFα, IL-2, GM-CSF, IL-12, IL-15, and others have shown positive results in several studies [63]. Dr. Zajakina concluded her talk by reporting the results of a study exploring the immunomodulatory and antitumor effects of an alphavirus (replication-deficient Semliki Forest virus (SFV)) vector expressing interferon-gamma (IFNγ). The study demonstrated that the SFV vector expressing IFNγ inhibited tumor growth, increased the number of tumor-infiltrating CD4 and CD8 T cells, and reduced T regulatory cells [64]. The talk concluded that viral vectors offer diverse immunotherapeutic opportunities and should be studied further for future application in cancer therapy.

Next, one of the early career researchers—Dr. **Yufei Xia** (Institute of Process Engineering, Beijing, China) described a new vaccine delivery system based on engineered Pickering emulsions. Vaccine delivery has become an important way to increase the overall vaccine efficacy by increasing the delivery and uptake of the target antigen and adjuvant, often simultaneously functioning as an immune stimulator. Dr. Xia reported the potential advantages of particle-stabilized emulsions—Pickering emulsions as vaccine delivery systems. Pickering emulsions consist of densely packed nanoparticles around an oily core. The outer particulate layer offers a large surface for antigen and other targets binding and subsequent interaction with cells, and the oily core ensures the emulsion’s softness and pliability, allowing it to pass in-between cell gaps. Current study by Xia et al. demonstrated that the use of the Pickering emulsion adjuvant system (PPAS) boosts cellular uptake. In vivo experiments found that the use of PPAS boosted uptake by infiltrated antigen-presenting cells, increased MHC class I presentation, and enhanced antigen accumulation in draining lymph nodes, thus activating lymph-node-residing immune cells. The study also found that using PPAS instead of solid particle vaccines stimulated their prophylactic and therapeutic effects in the case of influenza and the therapeutic melanoma vaccine [65]. In another study exploring the physiochemical properties of such emulsions, it was demonstrated that the softness and deformability enhance the biodistribution achieving faster and higher accumulation of antigens in lymph nodes and subsequent higher lymph-node-residing T-cell activation, as well as increases therapeutic effects in tumor-bearing mice [66]. A study combining the commonly used adjuvant alum with Pickering emulsions as a novel COVID-19 vaccine strategy demonstrated that the alum-stabilized Pickering emulsion (PAPE) absorbs large quantities of SARS-CoV-2 antigens, is readily up-taken by dendritic cells, produces higher antigen-specific antibody titers and induces higher T-cell activation [67]. The talk concluded that Pickering emulsions’ softness and other properties that affect vaccine uptake and efficacy make such formulations promising strategies for future vaccine delivery.

Prof. **Jorma Hinkula** (Department of Clinical and Experimental Medicine, Linköping University, Linköping, Sweden) presented the study of the gold nanoparticle adjuvants to be used in vaccines against viral infectious diseases affecting the respiratory tract. Gold nanoparticles (GNPs) activity as an adjuvant was explored in vitro and preclinical settings. GNPs are chemically inert and have shown minimal levels of cytotoxicity. Furthermore, their metallic properties allow them to interact with various functional groups and bind targets with high affinity. GNPs possess adjuvant properties, and the size and shape of the nanoparticles influence their impact on the immune system [68]. A previous study demonstrated that these ultrasmall GNPs could be synthesized in specific sizes with high precision and are highly bio- and immunocompatible [69]. In the reported study, BALB/c mice were intranasally and subcutaneously immunized with GNP vaccine candidates (10 nm and 40 nm particles) coupled with influenza rHA and M2e-peptides and SARS-CoV-2 S1 antigens. The study found that in mice who had received the gold-adjuvanted vaccine, significantly stronger humoral immune responses, especially against the HA and S1 antigens, could be observed. The highest serum IgG titers were observed in mice who were intranasally immunized.

The intranasal GNP administration also induced mucosal anti-HA and anti-S1 IgA responses in the respiratory tract. Furthermore, experiments evaluating the stability of the vaccine formulation found that GNP influenza rHA/SARS-CoV-2 S1 vaccine candidates could induce neutralizing titers of serum antibodies and IgA even after 6-month storage at room temperature. Finally, both intranasal and subcutaneous vaccination-induced protection against influenza A challenge.

The last study of the session about HBV core antigen (HBc) as a VLP-based platform for the development of SARS-CoV-2 vaccine prototypes was presented by one of the early career researchers Ms. **Anastasia Nepryakhina** (Latvian Biomedical Research and Study Centre, Riga, Latvia). Previous studies demonstrated that HBc of HBV genotype G has several advantages over other HBV genotypes—it can be obtained with higher outcomes and purity in *E. coli* expression systems and has high stability and packaging potential [70]. Based on this, the reported study aimed to create new VLPs based on HBc of genotype G to present SARS-CoV-2 spike and nucleocapsid epitopes. SARS-CoV-2 epitopes were selected from the original Wuhan strain, the Delta and Omicron variants. The obtained epitopes were inserted in the HBc C-terminus, and competent *E. coli* were transfected. After expression optimization and VLP purification by gel filtration and chromatography, the HBc VLPs presenting SARS-CoV-2 epitopes were acquired and confirmed by electron microscopy. These novel VLPs could serve as the basis of new SARS-CoV-2 vaccine candidates.

## 8. Sessions on Research Impact, Implementation and Funding

Drs. **Charlotta Lindquist** and **Patrik Blomquist** from Karolinska Intitutet (Sweden) gave talks highlighting how to achieve academic research with a higher impact and the commercial implications of research-driven innovations. Dr. **Ying Zhao** (Karolinska Institutet, Sweden) gave the detailed talk on the ABC of preparing a successful grant application. This talk gave valuable and detailed advice for all stages of the grant application process—from finding the correct calls to writing each section.

Two speakers from Latvia gave talks focusing more on research funding. First, Dr. **Uldis Berkis** (Ministry of Education and Science of the Republic of Latvia & Riga Stradins University, Riga, Latvia) highlighted new European Union funding opportunities for medical research. Second, Dr. **Janis Ancans** (Horizon Europe National Contact Point, Latvian Council of Science, Riga, Latvia) discussed funding opportunities for vaccine research and development in the European Union.

## 9. Conclusions and Remarks

The “Vaccines and Vaccination During and Post COVID Pandemics” online conference ended with the nomination of the winners of the **Early Career Researchers Contest**. The 1st place was awarded to Dr. Massimiliano Bissa (National Cancer Institute, Bethesda, Rockville, MD, USA), the 2nd place to Andrea Cerasuolo (Istituto Nazionale Tumori IRCCS Fond. Pascale, Naples, Italy), and the 3rd place was shared between two young researchers from the Riga Stradins University, Latvia, Zane Lucāne and Yuliya Raukach. All winners were granted monetary awards, and winners of the 1st and 2nd places also received vouchers for free publication in VACCINES.

The Conference allowed the presentation and the review of all aspects of modern development and production of vaccines spanning from the prevention of the current acute COVID-19 pandemic to therapeutic vaccines for chronic infections and their related cancers. The relevance and the current impact of such topics were highlighted by the speakers’ high quality and the attendees’ active participation.

## Data Availability

Not applicable.

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
