# Peer review of "Proceedings of the Online Conference “Vaccines and Vaccination during and Post COVID Pandemics” (7–9 December 2022)"

_vaccines, 2023, doi:10.3390/vaccines11071175_

Round 1

Reviewer 1 Report

The conference report on the online conference 'Vaccines and Vaccination During and Post COVID Pandemics' is very well written and summarizes the contributions to this meeting in a concise, yet highly comprehensible way. I have a few suggestions for further enhancing understandability. 

L10 and L27: Should be hesitancy not hesitance.

L139: Please provide some more information about common variable immunodeficiency and how it impacts antibody responses in general, e.g. how much lower are antibody levels in these patients?

L187: Please explain briefly what PrEP holds in.

L212: Do you mean: This, together with the fact that cases of spontaneous HIV remission are rare even in early treated children, ..?

L217: Should be 'HIV DNA vaccine'

L223: Can any information be given on when the study is expected to start?

L237: Should be 'antibody dependent cellular cytotoxicity'

L246: This part is difficult to understand, consider rephrasing.

L249: Maybe 'could affect the vaccine-evoked responses' ?

L260: Not clear what is meant by 'vaccine efficacy' here. Is it protection?

L270: herd immunity effects?

L285: Can something be said about the prevalence of HPV-associated tonsillar and base of tongue tumors in males and females?

L484: please explain the term heteroclitic

L713: The type and role of oligonucleotides here is not clear.

L884ff: This paragraph contains numerous language and typing errors, please check. 

The quality of English language is overall very good. Yet, the manuscript should be checked for mistakes with the use or absence of articles (the, a/an). 

Author Response

Dear Reviewer,

We appreciate the time You dedicated to review our manuscript and Your valuable feedback. We have revised the manuscript accordingly:

  • We have corrected the language and typing errors;
  • Per the comments in the review report, we have added extra information that will aid the understandability of the manuscript:
    • Short description of the primary antibody deficiencies with references;
    • Brief explanations of PrEp and heteroclitic peptides;
    • Added the statement that the HVRICCANE study is currently recruiting participants;
    • A reference to the gender disparity in head and neck cancer prevalence;

Thank You once again for Your thorough review and for guiding us toward enhancing the quality of our manuscript. We hope that the revised manuscript adequately addresses Your concerns. 

Best regards, 

The authors

Reviewer 2 Report

Dear Authors,

Nice report. Why does the title say 'pandemics'? Shouldn't it be 'pandemic'? Perhaps the introduction needs some rationale on why such conference.

I made some suggestions in the file attached with sections highlighted and balloons. Some references are missing (CCHF, Hepatic Cancer) and need to be added. It is also curious for HPV there is no reference to the important and abundant work (Costa Rica, India) on the virtue of single dose HPV vaccination with proof of long term protection against pre-clinical and clinical cervix cancer. You may want to discuss with the presenter.

Re. CCHF, please do add the work of Prof. Ali Mirazmi at KI and his references. He showed 100% protection after DNA vaccination in NHPs and 100% protection in mice after mRNA vaccination. Of importance, CCHF's geographic expansion (including in Sweden) is linked to climate change.

Typos here and there and large spaces between some words, or no space between word and following reference in [X].

Author Response

Dear Reviewer,

We appreciate the time You dedicated to review our manuscript and Your valuable feedback. We have revised the manuscript according to the issues You raised:

  • We have addressed the language, typing, and spacing errors;
  • We have added a paragraph to the “Introduction” section to provide some rationale for the organization of the conference;
  • We have added references to the statements You highlighted (where possible, since in some cases, the text referred to the presented studies, which in some cases were not yet published);
  • We have added the references you recommended (e.g. the studies on DNA vaccination against CCHFV from Karolinska Institutet).

Additionally, we would like to address some of the issues You raised:

  • The word “Pandemics” was used in the official title of the conference (present in banners, abstract book, and the conference website), which is why we did not revise it in the manuscript;
  • The studies on HPV presented at the conference mainly focused on results obtained from countries the researchers resided in, thus even though the studies You highlighted are important, they were not mentioned;

Thank You once again for Your thorough review and for guiding us toward enhancing the quality of our manuscript. We hope that the revised manuscript adequately addresses Your concerns.

Best regards, 

The authors

Reviewer 3 Report

In the Conference Report “Proceedings of the "Vaccines and Vaccination During and Post 2 Covid Pandemics" online conference (December 7 – 9, 2022)” written by Drs Sokolovska, Isaguliants and Buonaguro, the authors summarize the presentations and discussions presented during the VAC&VAC2022 conference held on the platform of Riga 13 Stradins University, Riga, Latvia.

The report covers all the topics that were discussed during the conference, following the same organization that was adopted during the VAC&VAC2022. The authors summarized the findings and updates presented by each presenter covering all the main aspects presented by the researchers and reporting them accurately and concisely.

For these reasons the report is suitable for publication and of great help for those interested in a summary of the VAC&VAC2022 conference.

Finally, the report is well written and fully understandable, it follows a temporal logic that allows the readers to remain engaged and interested in the summary.

Please note that there are some double-spaces and not all the names of presenters are reported in bold.

Author Response

Dear Reviewer,

Thank You for taking the time to review our manuscript titled “Proceedings of the "Vaccines and Vaccination During and Post 2 Covid Pandemics" online conference (December 7 – 9, 2022)”. We appreciate Your assessment and positive feedback.

Best regards, 

The authors